# Stereotactic Ablative Radiotherapy for Oligometastatic Hepatocellular Carcinoma: A Multi-Institutional Retrospective Study (KROG 20-04)

**DOI:** 10.3390/cancers14235848

**Published:** 2022-11-27

**Authors:** Tae Hyung Kim, Taek-Keun Nam, Sang Min Yoon, Tae Hyun Kim, Young Min Choi, Jinsil Seong

**Affiliations:** 1Departments of Radiation Oncology, Yonsei Cancer Center, Yonsei University College of Medicine, Seoul 03722, Republic of Korea; 2Department of Radiation Oncology, Nowon Eulji Medical Center, Eulji University School of Medicine, Seoul 01830, Republic of Korea; 3Department of Radiation Oncology, Chonnam National University Medical School, Gwangju 61469, Republic of Korea; 4Department of Radiation Oncology, Asan Liver Center, Asan Medical Center, University of Ulsan College of Medicine, Seoul 05505, Republic of Korea; 5Center for Proton Therapy, National Cancer Center, Goyang 10408, Republic of Korea; 6Department of Radiation Oncology, Dong-A University College of Medicine, Busan 49201, Republic of Korea

**Keywords:** hepatocellular carcinoma, radiotherapy, oligometastasis, overall survival

## Abstract

**Simple Summary:**

The concept of oligometastasis has not been widely shared in the hepatocellular carcinoma (HCC). This study is a multi-institutional retrospective study. We reviewed the data of patients who received SABR for HCC. The results showed that SABR for all metastatic lesions appeared to be an effective and safe treatment for oligometastatic HCC. The median OS for all patients was 16 months, with a 2-year OS rate of 40%. Long-term survival is expected in patients with good performance status and liver function.

**Abstract:**

We investigated the clinical efficacy of stereotactic ablative radiotherapy (SABR) in patients with oligometastatic hepatocellular carcinoma (HCC). The inclusion criteria were patients receiving definitive treatment for HCC with 1–5 metastatic lesions, <3 metastases in a single organ and receiving radiotherapy with fraction doses ≥6 Gy. A total of 100 patients with 121 metastatic lesions were reviewed. The most common site of metastasis was the bones (40%), followed by the lungs (38%). Systemic therapy was administered to 71% of patients. With a median follow-up of 13 months, the median overall survival (OS) was 16 months. The 2-year OS rate was 40%. The prognostic factors in univariate analysis were performance status, Child–Pugh class, primary HCC status, and time interval of metastasis. Performance status and Child–Pugh class remained in multivariate analysis. OS differed significantly depending on the number of prognostic factors: 46 months in patients with both factors (Group 1), 13 months with one factor (Group 2), and 6 months with no risk factor (Group 3) (*p* < 0.001). Nine patients experienced grade 1 radiation pneumonitis. Given its efficacy and safety, SABR deserves active consideration in the treatment of oligometastatic HCC.

## 1. Introduction

Hepatocellular carcinoma (HCC) is the third-leading cause of cancer-related deaths worldwide, with nearly 780,000 deaths annually [1]. However, only 30–40% of patients are diagnosed at early stages, which are amenable to potentially curative treatment. In patients with advanced disease, treatment is mostly aimed at palliation, despite recent developments in systemic therapies [2,3,4]. The treatment of patients with metastatic solid tumours has been based on systemic therapies that aim to delay progression and extend life rather than on complete disease eradication [5]. The oligometastatic paradigm, formally defined in the 1990s [6], suggests that metastatic disease is not widespread and may be amenable to a curative treatment approach in some patients. With recent advances in imaging techniques and cancer-specific imaging strategies, more patients are being diagnosed with oligometastatic cancer. Stereotactic ablative body radiotherapy (SABR) can stimulate the systemic immune response through tumour cell death and immune system recognition [7]. Thus, interest in treating oligometastatic diseases is increasing with improvements in systemic therapy and immunotherapy.

Recently, the SABR-COMET randomised phase 2 clinical was conducted to improve the oncological outcomes of SABR [8]. Long-term follow-up of the SABR-COMET revealed that SABR in all metastatic lesions was associated with improved overall and progression-free survival (OS and PFS, respectively) [9].

However, in HCC, the concept of oligometastasis has not been widely shared in the liver cancer community. SABR to metastatic lesions is often attempted according to physicians’ preferences. Consequently, the clinical efficacy of SABR in oligometastasis of HCC has not yet been fully explored. Thus, this study investigated the clinical efficacy of SABR in patients with oligometastatic HCC.

## 2. Materials and Methods

### 2.1. Study Population

The Korean Radiation Oncology Group (KROG) is a research coalition of radiation oncologists under the Korean Society of Radiation Oncology. This study included five institutions belonging to and endorsed by the KROG. After receiving approval from the institutional review boards of each institution, we reviewed the data of patients who received SABR for HCC between January 2000 and December 2019. This study was approved by the Severance Hospital institutional review board (IRB #4-2021-1101). Because this study was retrospective, the need for written informed consent was waived. The inclusion criteria were age ≥18 years, primary HCC that had been treated definitively before enrolment, metastatic disease established by imaging studies, metastatic lesions with a maximum of three metastases in any one organ with no more than five metastases in total, and metastatic disease treated with SABR with a fraction dose ≥6 Gy. Radiologic imaging studies included computed tomography (CT), magnetic resonance imaging (MRI), whole-body bone scan (WBBS), and position emission tomography (PET). Patients with a poor performance status (Eastern Cooperative Oncology Group (ECOG) score of 3–4) were excluded. Patients who received other metastasis-directed therapy were excluded. The treatment flow diagram is shown in Figure 1.

Controlled primary HCC was defined as a primary tumour treated definitively at least 3 months before SABR, with no sign of disease progression on subsequent imaging studies. The time interval between primary HCC treatment and the diagnosis of metastasis was calculated and categorised as synchronous (≤6 months) or metachronous (>6 months). Performance status was graded at the time of treatment by ECOG score. 

Patients administered systemic therapy before and after SABR were included. Patients received systemic therapy consisting of either a tyrosine kinase inhibitor or chemotherapeutic agent.

Treatment-related toxicities were monitored at least once weekly during treatment, and more often if clinically indicated. Treatment-related toxicities were graded according to the Common Toxicity Criteria for Adverse Events version 4.0 [10]. Acute toxicities were defined as adverse events during radiotherapy (RT) and late toxicities as adverse events 1 month after RT and were assessed from patient records.

### 2.2. Radiotherapy

All patients received SABR at all sites of metastatic disease to achieve maximum local control while diminishing potential RT-related toxicities. The SABR protocol used in each institution was widely accepted, but in general, the doses ranged from 30 to 60 Gy in 3–8 fractions, depending on the tumour size and location. In all situations, dose constraints to normal tissue did not exceed even though the dose to all or part of the target had to be reduced. The treatments were delivered using static beams (either three-dimensional-conformal or intensity-modulated RT) or rotational therapy (volumetric-modulated arc therapy or tomotherapy). Quality assurance of the SABR and institutional peer review was not conducted. 

### 2.3. Response Measurement and Statistical Analysis

Patients were interviewed by a physician before the start of SABR, 2 weeks after SABR, 2 months after SABR, and every 3 months thereafter for 1 year to measure treatment response and toxicities. OS was calculated from the date of RT initiation to the date of death or the last follow-up. Progression was defined as disease progression at any site or death. Treatment response was assessed using the Response Evaluation Criteria for Solid Tumors version 1.1 [11]. Complete response was defined as the disappearance of all target lesions. Partial response was defined as a decrease in the sum of the longest diameters of the target lesions of ≥30%, using the corresponding baseline values as reference values. Stable disease was defined as shrinkage not meeting the criteria for partial response or progressive disease, with the smallest sum of the longest diameter since treatment initiation as the reference. Progressive disease was defined as either an increase of ≥20% in the sum of the longest diameters of the target lesions, with the corresponding smallest sum recorded since the treatment initiation used as the reference, or the appearance of one or more new lesions. The failure patterns were determined using the site of the first failure. Failure was defined as the reappearance of a lesion that showed a complete response or the appearance of any new lesion. Local failure was defined as a failure-occurring lesion that was treated with SABR. In case of metastatic bone lesions, local failure was defined as increased hot uptake observed by WBBS or progression of bone destruction detected by CT or MRI. Distant failure involved any site beyond the lesion that was treated with SABR.

The differences in characteristics and toxicities were compared using chi-square tests. The Kaplan–Meier method was used to calculate the OS and PFS, with the differences between the curves analysed using log-rank tests. A Cox proportional hazards model was used to assess the association of variables with survival and to calculate hazard ratios. The multivariable analysis included factors that showed statistical significance in univariate analysis (*p* < 0.1). The statistical analyses were conducted using IBM SPSS, version 25.0 (IBM Corp., Armonk, NY, USA), with *p* < 0.05 considered statistically significant.

## 3. Results

This section may be divided by subheadings. It provides a concise and precise description of the experimental results and their interpretation, as well as the experimental conclusions that can be drawn.

### 3.1. Patient

The median age of the patients was 62 years (range, 40–83 years), and the male-to-female ratio was 8:2. Most of the patients (83%) were Child–Pugh class A, and 58% of the patients had controlled primary HCC (Table 1).

A group of 17 patients were diagnosed with synchronous oligometastasis, while 83 patients were diagnosed with metachronous oligometastasis. In this study, 85%, 9%, and 6% of patients had one, two, and three metastatic lesions, respectively. The most common site of metastasis was the bone (40%), followed by the lungs (38%), abdominal lymph nodes (LNs) (11%), adrenal glands (5%), and others (5%). Systemic therapy was administered to 71% of patients. Most patients were treated with tyrosine kinase inhibitors (59%) and systemic chemotherapy (12%). A total of 12 patients received systemic therapy during SABR, and 32 (45%) patients received systemic therapy after SABR (Table 2).

In patients with oligometastatic bone lesions (*n* = 55), the most common metastatic site was the L spine (33%), followed by the T spine (27%) and pelvic bone (18%). The most frequently used RT dose scheme was 60 Gy in four fractions (Appendix A). In patients with oligometastatic lung lesions (*n* = 42), the most common sites of metastases were the left and right lower lobes (29% and 29%), followed by the left upper lobe (19%) and right upper lobe (14%). The most frequently used RT dose scheme was 60 Gy in 4 fractions (Appendix A).

### 3.2. Survival

With a median follow-up of 13 months (range, 2–92 months), the median PFS and OS were 4 and 16 months, respectively (Figure 2). The 2-year OS rate was 40%. The median OS by metastatic site was not reached in the lung and 46 and 22 months in the adrenal gland and abdominal LN, respectively.

In univariate analysis, performance status, Child–Pugh class, primary HCC status, and time interval of metastasis were significant factors for OS (Figure 3). In multivariate analysis, performance status (*p* = 0.040) and Child–Pugh class (*p* = 0.018) were significant factors for OS (Table 3).

To determine the best prognostic subgroup, patients were divided into three groups according to the risk factors for OS, including performance status and Child–Pugh class. The favourable risk factors were ECOG Performance Status 0 and Child–Pugh class A. OS was significantly different depending on the number of prognostic factors: 46 months in patients with both factors (Group 1), 13 months with one factor (Group 2), and 6 months with no factor (Group 3) (*p*<0.001) (Figure 4).

### 3.3. Patterns of Failure and Toxicity

A total of 84 patients experienced treatment failure. The most common pattern was distant failure (*n* = 69, 82%). The sites of distant failure were the liver (*n* = 40), contralateral lung (*n* = 11), bone (*n* = 10), abdominal LN (*n* = 6), brain (*n* = 1), and rectus abdominis muscle (*n* = 1). Patients with failure at the liver (*n* = 40) received salvage treatment consisting of liver-directed therapy (trans-arterial chemoembolization, *n* = 16), systemic therapy (*n* = 11), RT (*n* = 6), and best supportive care (*n* = 7). Patients with failure at the lungs (*n* = 11) received salvage treatment consisting of systemic therapy (*n* = 7), RT (*n* = 3), and surgery (*n* = 1). Patients with failure at the bones (*n* = 10) received salvage treatment consisting of RT (*n* = 7) and systemic therapy (*n* = 3). Patients with failure at the abdominal LN (*n* = 6) received salvage treatment consisting of RT (*n* = 3), systemic therapy (*n* = 2), and best supportive care (*n* = 1).

In this study, 16% of patients experienced local failure in the bone (*n* = 9), lung (*n* = 6), and lymph nodes (*n* = 1). Patients with local failure after SABR for metastatic bone lesions received salvage treatment with re-irradiation (*n* = 5), systemic therapy (*n* = 2), surgery (*n* = 1), and best supportive care (*n* = 1). Patients with local failure after SABR for metastatic lung lesions received salvage treatment with re-irradiation (*n* = 3), systemic therapy (*n* = 2), and best supportive care (*n* = 1). One patient who experienced local failure after SABR to the abdominal LN received systemic therapy.

No patients experienced acute treatment-related toxicities, while nine patients experienced grade 1 radiation pneumonitis. Among them, eight patients received SABR for lung lesions, and one patient received SABR for peritoneal seeding close to the lower lung. One patient underwent SABR for two lung lesions. A group of 5 patients received SABR at a dose of 60 Gy in four fractions

## 4. Discussion

The results of the current study showed that SABR for all metastatic lesions appeared to be an effective and safe treatment for oligometastatic HCC. The median OS for all patients was 16 months, with a 2-year OS rate of 40%. The median OS for patients with a good performance status and good liver function was 46 months. No patients experienced acute treatment-related toxicities, while nine patients experienced grade 1 radiation pneumonitis.

Oligometastasis, a concept first described in 1995 by Hellman and Weichselbaum, is a mixture of several concepts [12], including synchronous/metachronous oligometastasis, repeat/induced oligometastasis, and oligorecurrence/oligopersistance. As each concept refers to a different disease condition, it is difficult to simultaneously apply the same treatment protocol with oligometastasis. In addition, as the diagnosis of oligometastasis is based entirely on radiographic findings, the concept of oligometastasis may evolve with advances in imaging technology. For example, prostate-specific membrane antigen-positron emission tomography scans can be used to detect subcentimeter metastases [13]. However, oligometastasis is not yet well defined in HCC compared to other major solid cancers.

In our study, univariate analysis showed a worse prognosis in patients with synchronous oligometastasis compared with those with metachronous oligometastasis. Synchronous oligometastatic disease was associated with a worse prognosis compared to metachronous oligometastatic disease [14]. Additionally, patients with controlled primary HCC had better survival outcomes than those with uncontrolled primary HCC. The causes of death in patients with HCC included uncontrolled tumour growth and liver failure [15]. In the present study, additional analysis with the two risk factors that were significant in multivariate analysis—performance status and Child–Pugh class—showed a median OS for patients with good performance status and good liver function of 46 months, compared with 13 and 6 months, respectively, for patients with one or no factors. Therefore, local ablative therapy for oligometastasis should consider the prognostic factors analysed above and should be discussed by a multidisciplinary team.

The concept of eradicating oligometastasis using local ablative therapy has been proven through prospective trials in several cancers. In addition to the widely known SABR-COMET trial [9], the ORIOLE trial in prostate cancer [16], a phase II randomised trial in non-small cell lung cancer [17], and a randomised trial in colorectal liver metastases proved this concept [18]. However, for HCC, only retrospective studies conducted in a single institution have been performed [19,20]. This is valuable because they were conducted as multi-institutional studies of HCC.

There remain no universally accepted dose prescriptions. We identified various dose prescriptions among the multiple institutions in this study. While a recent pooled analysis showed no clear evidence supporting a dose-response relationship [21], one study showed that a biologically effective dose ≥100 Gy was associated with improved local tumour progression [22]. In the present study, as SABR was administered for each of the metastatic diseases in various organs, the dose-response could not be studied. Considering Norman Coleman’s statement that ‘Radiation is a different drug at different doses and fractionation schedules’, SABR dose and fractionation might be linked to the local and distant tumour response. In the era of immunotherapy, since reducing the number of fractions of RT could reduce RT-induced lymphopenia [23,24], SABR should be considered an important treatment option. Therefore, we urge the development of a consensus guideline for SABR in HCC endorsed by an international working group in a prospective clinical trial. 

The present study had several limitations. First, this study was a retrospective analysis, and acute and chronic toxicities may have been under-evaluated. In addition, the study included patients from several institutions using various SABR doses and fractionation schemes; thus, the results should be interpreted with caution. Finally, this study was performed over a long period between 2000 and 2019; hence, there might have been changes in diagnostic tools, systemic therapy, and RT techniques. Therefore, future prospective studies are needed to determine the efficacy of SABR for the treatment of oligometastatic HCC.

## 5. Conclusions

With SABR, long-term survival can be expected in patients with good performance status and good liver function. Given its efficacy and safety, SABR deserves active consideration in the treatment of oligometastatic HCC. Further prospective trials are needed to prove the benefits of SABR for oligometastatic HCC.

## Figures and Tables

**Figure 1 cancers-14-05848-f001:**
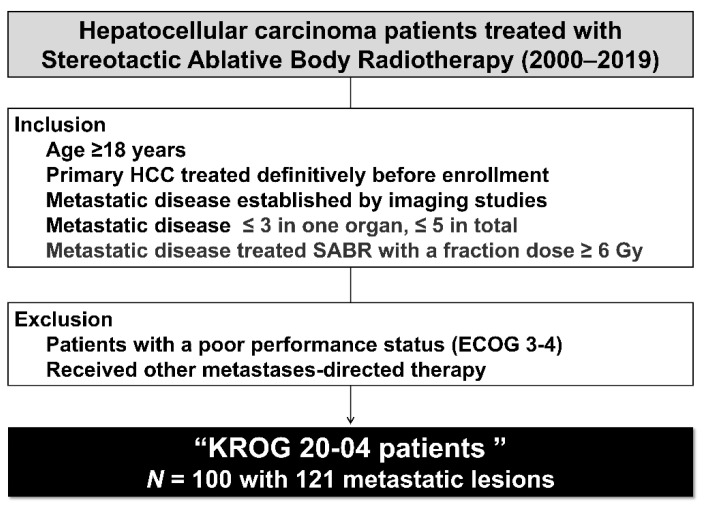
Study flow diagram.

**Figure 2 cancers-14-05848-f002:**
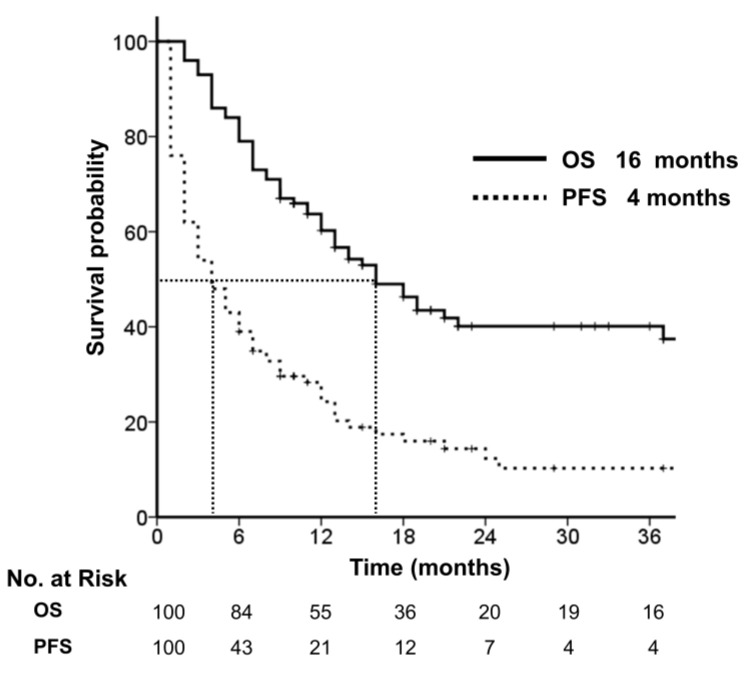
Overall and progression-free survival.

**Figure 3 cancers-14-05848-f003:**
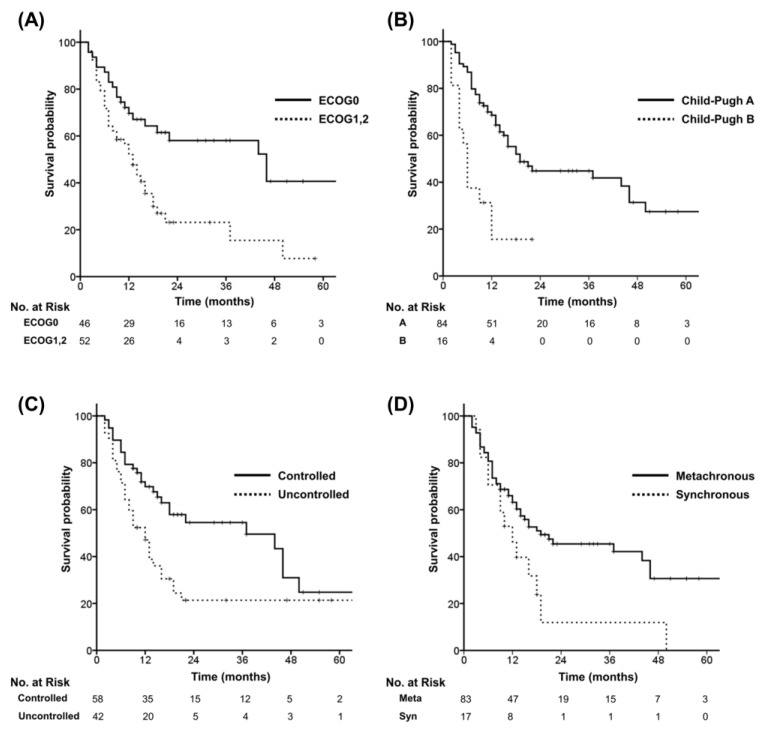
Overall survival according to risk factors identified in univariate analysis. (**A**) Performance status. (**B**) Child–Pugh Classification. (**C**) Primary tumour status. (**D**) Time interval of metastasis.

**Figure 4 cancers-14-05848-f004:**
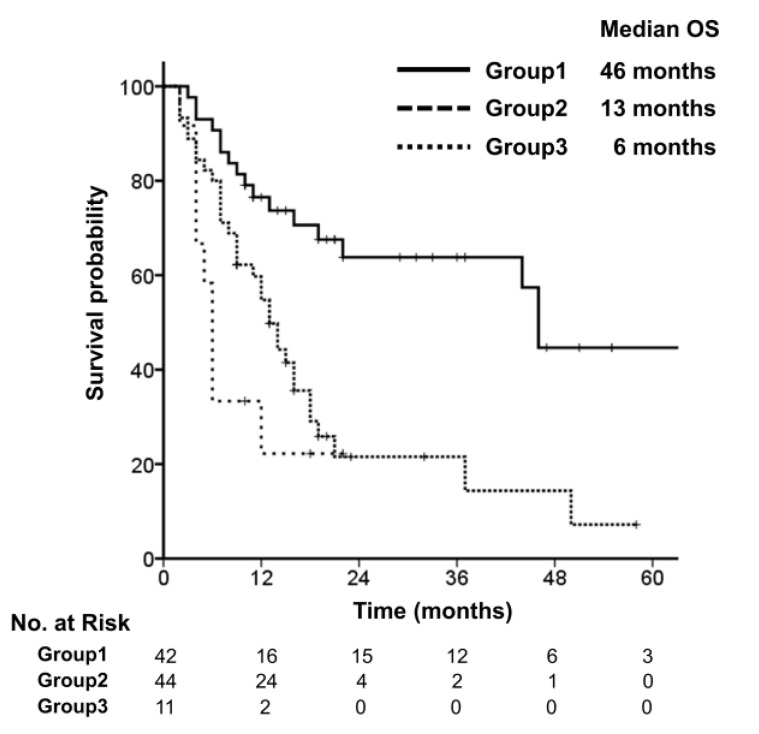
Overall survival, stratified by groups with favourable risk factors.

**Table 1 cancers-14-05848-t001:** Patient characteristics (*n* = 100).

Variables	*n*
Age (in years)	
Median	62
Range	40–83
Sex	
Male	82
Female	18
Performance status	
ECOG0	47
ECOG1	50
ECOG2	3
Etiology	
HBV	66
HCV	11
NBNC	23
Child-Pugh class	
A	83
B	17
Child-Pugh score	
CPS 5	73
CPS 6	10
CPS 7	17
Primary HCC	
Controlled	58
Uncontrolled	42

ECOG PS, Eastern Cooperative Oncology Group Performance Status; HBV, hepatitis B virus; HCV, hepatitis C virus; NBNC, non-HBV/HCV; CPS, Child–Pugh score; HCC, hepatocellular carcinoma.

**Table 2 cancers-14-05848-t002:** Metastasis characteristics and treatment details.

Variables	*n*
Metastasis characteristics	
Time interval	
Synchronous oligometastasis	17
Metachronous oligometastasis	83
Number of metastasis	
1 lesion	85
2 lesions	9
3 lesions	6
Site of metastases	
Bone	40
Lung	38
Abdominal lymph node	11
Adrenal gland	5
Others	5
Combined lung & bone	1
Details of treatment	
Systemic therapy	
No	29
Tyrosine kinase inhibitor	59
Chemotherapy	12
Timing of systemic therapy	
Pre-SABR	27
During-SABR	12
Post-SABR	32

SABR, stereotactic ablative body radiotherapy.

**Table 3 cancers-14-05848-t003:** Prognostic factors for overall survival.

Variables	Univariate				Multivariate	
	HR	95% CI	*p* Value	HR	95% CI	*p* Value
Age (Continuous, year)	1.012	0.989–1.035	0.306			
Sex (Female vs. Male)	0.832	0.419–1.653	0.600			
Performance (ECOG0 vs.ECOG1&2)	2.375	1.375–4.103	0.002	1.840	1.028–3.295	0.040
Child-Pugh class (A vs. B)	3.344	1.767–6.327	<0.001	2.277	1.153–4.495	0.018
Etiology (HBV vs. HCV)	1.326	0.612–2.872	0.475			
Primary HCC (Controlled vs. Uncontrolled)	1.957	1.171–3.268	0.010	1.437	0.831–2.484	0.195
Time interval (Syn vs. Metachronous)	0.525	0.287-0.962	0.037	0.761	0.405-1.430	0.396
Number of metastases (1 vs. 2&3)	1.220	0.632–2.357	0.553			
Systemic therapy (No vs.TKI)	1.285	0.715–2.310	0.401			
Systemic therapy (No vs. chemotherapy)	1.289	0.533–3.114	0.573			

ECOG PS, Eastern Cooperative Oncology Group Performance Status; HBV, hepatitis B virus; HCV, hepatitis C virus; NBNC, non-HBV/HCV; HCC, hepatocellular carcinoma; Syn, synchronous; TKI, tyrosine kinase inhibitor.

## Data Availability

The datasets generated and/or analyzed during the current study are not publicly available due to risk of personal information leakage but are available from the corresponding author on reasonable request.

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
