# Peer review of "Stereotactic Ablative Radiotherapy for Oligometastatic Hepatocellular Carcinoma: A Multi-Institutional Retrospective Study (KROG 20-04)"

_cancers, 2022, doi:10.3390/cancers14235848_

Round 1
Reviewer 1 Report
This is a retrospective study looking at outcomes of patients with oligometastatic liver cancer treated with SABR. I think the results of your study show that this is a safe approach. I do not feel like this data is adequate enough to call it an effective approach. There may be a suggestion of efficacy, but I definitely not conclusive. The 3 groups you describe that have different OS based on PS and CP- class are well established prognostic factors in liver cancer, so i do not find this very novel.
Reviewer 2 Report
The manucript presents results of multicentric retrospective study on effect of stereotactic ablative radiotherapy (SABR) in patients with oligometastatic hepatocellular carcinoma (HCC). The topic is important, because the clinical efficacy of SABR in this indication has not been fully explored. A total of 100 patients treated in the period between 2000 - 2019 were included. The median OS for all patients was 16 months and the 2-year OS rate was 40%. Results were better for patients with good performance status and good liver function. The doses ranged from 30 to 60 Gy in 3 - 8 fractions.
Study procedures including analyses of influence of synchronous vs metachronous oligometastases, number and site of metastases and role of performance status and liver function on the results are adequate and well described.
The limitation of the study are mentioned in the discussion.
In the general, the article brings new informations, it is carefully prepared and well written.
Only one question to the authors: was it possible to observe any possible abscopal effect of SABR?
Reviewer 3 Report
This manuscript entitled Stereotactic ablative radiotherapy (SABR) for oligometastatic hepatocellular carcinoma: a multi-institutional retrospective study (KROG 20-40) reviewed the data of patients who received SABR in the period from year 2000 to 2019 in five medical center. The results showed that all the 121 metastatic lesions in 100 patients appeared to be effective and safe by SABR with fraction doses >6 Gy. The median overall survival for all patients was 16 months, with a 2-year OS rate of 40%. The prognostic factors in nunvariate analysis were performance status, Child-Pugh class, primary HCC status and time interval of metastasis. This report concluded that SABR was an effective and safe treatment for oligometastatic HCC and long-term survival is expected in patients with good performance status and liver function.
There are some critiques and suggestions:
1. Line 75, Patients with a poor performance status score of 3-4 and a life expectancy of <6 months were excluded. However, this is a retrospective study and the survival time was recorded in the medical chart. How to exclude the patients with a life expectancy of <6 months. Figure 3 showed the overall survival, stratified by groups with favourable risk factors and demonstrated the median OS in Group3 was 6 months. The authors need to clarify this data.
2.Suggest the authors to present the followup plan after SABR in section 2.3 Response measurement and statistical analysis.
3. Suggest the authors to draw a study schema or flowchart to show a whole picture in Materials and Methods
Round 2
Reviewer 3 Report
The authors well response the critiques and suggestions point-by-point. There are no additional comments.